

# Evaluation of standard imaging techniques and volumetric preservation of nervous tissue in genetically identical offspring of the crayfish *Procambarus fallax* cf. *virginalis* (Marmorkrebs)

Emanuel S. Nischik and Jakob Krieger

Zoological Institute and Museum, Cytology and Evolutionary Biology, University of Greifswald, Greifswald, Germany

Corresponding author
Jakob Krieger,
jakob.krieger@uni-greifswald.de

## ABSTRACT

In the field of comparative neuroanatomy, a meaningful interspecific comparison demands quantitative data referring to method-specific artifacts. For evaluating the potential of state-of-the-art imaging techniques in arthropod neuroanatomy, micro-computed X-ray microscopy (μCT) and two different approaches using confocal laser-scanning microscopy (cLSM) were applied to obtain volumetric data of the brain and selected neuropils in *Procambarus fallax* forma *virginalis* (Crustacea, Malacostraca, Decapoda). The marbled crayfish *P. fallax* cf. *virginalis* features a parthogenetic reproduction generating genetically identical offspring from unfertilized eggs. Therefore, the studied organism provides ideal conditions for the comparative analysis of neuroanatomical imaging techniques and the effect of preceding sample preparations of nervous tissue. We found that wet scanning of whole animals conducted with μCT turned out to be the least disruptive method. However, in an additional experiment it was discovered that fixation in Bouin's solution, required for μCT scans, resulted in an average tissue shrinkage of 24% compared to freshly dissected and unfixed brains. The complete sample preparation using fixation in half-strength Karnovsky's solution of dissected brains led to an additional volume decrease of 12.5%, whereas the preparation using zinc-formaldehyde as fixative resulted in a shrinkage of 5% in comparison to the volumes obtained by μCT. By minimizing individual variability, at least for aquatic arthropods, this pioneer study aims for the inference of method-based conversion factors in the future, providing a valuable tool for reducing quantitative neuroanatomical data already published to a common denominator. However, volumetric deviations could be shown for all experimental protocols due to methodological noise and/or phenotypic plasticity among genetically identical individuals. MicroCT using undried tissue is an appropriate non-disruptive technique for allometry of arthropod brains since spatial organ relationships are conserved and tissue shrinkage is minimized. Collecting tissue-based shrinkage factors according to specific sample preparations might allow a better comparability of volumetric data from the literature, even if another technique was applied.

## INTRODUCTION

Micro-computed X-ray tomography and three-dimensional reconstruction of internal morphological structures have opened up new possibilities for analyzing the anatomy of nervous systems in intact specimens (reviewed in *Metscher, 2009*; *Sombke et al., 2015*). In addition to vertebrates, the application of X-ray microscopy of soft tissues has been demonstrated to be useful for a variety of metazoan taxa, such as cnidarians (*Holst et al., 2016*), nematomorphs (*Henne et al., 2016*), nematodes (*O'Sullivan et al., 2017*), polychaetes (*Dinley et al., 2010*; *Faulwetter et al., 2013*), mollusks (*Handschuh et al., 2013*), as well as arthropods (*Akkari, Enghoff & Metscher, 2015*; *Michalik et al., 2013*; *Sombke et al., 2015*; *Steinhoff et al., 2017*).

Whole-mount scanning using confocal laser-scanning microscopy (cLSM) (*Krieger et al., 2015*; *Ott & Elphick, 2003*; *Ott, 2008*) and especially micro-computed X-ray microscopy (μCT) are suitable methods for precisely capturing three-dimensional structures without the need of histological sections. In samples with various tissue structures densely packed together, most imaging techniques require elaborate contrasting- or staining procedures (*Metscher, 2009*). In most cases, the tissue must be fixed and dehydrated in advance. The fixation process ideally fulfills the function of counteracting the structural and morphological changes induced by decay, which begins immediately after death of an organism (*Lang, 2013*). The chosen fixative and its time of penetration, subsequent preparation and imaging technique crucially influence the preservation of the tissue's spatial morphology on which the accuracy of allometric analysis essentially depends. Ethanolic and aldehydic solutions containing formaldehyde (or its polymer paraformaldehyde (PFA)) or glutaraldehyde are the most widely used chemical fixatives in histology. Aldehydic fixatives protect proteins against denaturation by cross-linking them. No fixative preserves all structures alike, thus usually mixtures are applied (e.g., Bouin or Karnovsky) to compensate adverse effects of single components. Obviously, the preparation should aim to maintain an isotonic milieu for tissue preservation (reviewed in *Lang, 2013*). For μCT, contrast-enhancing, especially for the visualization of soft tissues (*Gignac et al., 2016*; *Holst et al., 2016*; *Metscher, 2009*; *Mizutani & Suzuki, 2012*), is usually realized by increasing adsorption of X-rays applying solutions containing iodine ($I_2$), osmium tetroxide ($OsO_4$), or phosphotungstic acid ($H_3PW_{12}O_{40}$) as was demonstrated by *Metscher (2009)*. The sample manipulation in preparation for tomographic imaging for both cLSM as well as μCT, generally increases the vulnerability to artifacts (*Buytaert et al., 2014*). Shrinkage artifacts, due to chemical treatment (*Lang, 2013*) or the scanning itself (*Gianoncelli et al., 2015*), are very common and apart from mechanical distortions (e.g., caused by bruises and cracks), cannot be fully eliminated even with careful handling (*Buytaert et al., 2014*). Furthermore, artifacts induced by radiation or chemicals might be occasionally neglected or greatly underestimated in comparative scientific reviews, which might lead to erroneous interpretations.

Since more than 100 years, neuroanatomists always aspired to make their findings quantifiable by inventing more and more ingenious morphometric and allometric methods (see *Hanström, 1926*; *Snell, 1892*). For a meaningful interspecific comparison, quantitative

data referring to method-specific artifacts could serve as rewarding tool. Increasing availability of neuroanatomical volumetric data, collected by the use of different methodological approaches of various arthropods (*Beltz et al., 2003*; *Grabe et al., 2015*; *Hanström, 1926*; *Schmidt, 2016*; *Sombke et al., 2015*; *Tuchina et al., 2015*), raises demand for standardizing these data a posteriori, which constitutes the emphasis of this study. For this purpose, three different well-established sample preparations including two tomographic imaging techniques (cLSM and μCT) were performed on *P. fallax* cf. *virginalis*. Well identifiable and quantifiable substructures in the crayfish brain such as the deutocerebral chemosensory neuropil and the (deutocerebral) accessory neuropil served as approximation for volume change of tissues. The parthenogenetic marbled crayfish *P. fallax* forma *virginalis* (*Martin et al., 2010*) provides for a rewarding model organism (review: *Harzsch, Krieger & Faulkes, 2015*) featuring a textbook example of a "typical" astacid brain. The uncomplicated husbandry and year-round parthenogenetic reproduction of the marbled crayfish enables replicable tests using high sample numbers (review: *Vogt, 2011*).

## MATERIALS AND METHODS

### Nomenclature

The neuroanatomical nomenclature used in this manuscript is based on *Sandeman et al. (1992)* and *Richter et al. (2010)* with some modifications adopted from *Harzsch & Hansson (2008)*, *Kenning & Harzsch (2013)*, and *Loesel et al. (2013)*. The term "oesophageal connective" and the corresponding abbreviation OC (British English) are maintained here for simplicity. The syncerebral brain mass excluding the lateral protocerebrum and visual neuropils of the eyestalks (see *Krieger et al., 2015*) is termed "central brain" throughout the text according to *Schmidt (2016)*. Although, sample preparations as well as imaging techniques according to the three protocols used vary in a vast of parameters that influence the resulting volume data, each of the three fixatives (zinc-formaldehyde (ZnFA) fixative, Bouin's as well as half-strength Karnovsky's solution) is used as a synonym for the entire protocol throughout the text.

### Animals

The specimens of *P. fallax* cf. *virginalis* used in our study were obtained from a commercial aquarium shop (Aquaristik-Langer GbR, https://aquaristik-langer.de/) and were kept together in a freshwater tank at the facilities of the University of Greifswald. For the experimental design, berried individuals were isolated from a community tank, to ensure that all larvae originated from a single individual, and hence were genetically identical. A total of 24 individuals were harvested for pre-tests directly after the juveniles left the maternal pleopods. At this developmental stage, the average body length (rostrum to pleon) measured about 5 mm and the juveniles started autonomous feeding. Morphometric measurements were performed on dead and undissected juveniles in order to ensure that individuals were of the same size. Morphometric analysis was conducted using a Nikon eclipse 90i microscope connected to a Nikon camera DS2-MBWc. For each specimen, carapace length and eyestalk width were measured by using the software NIS-Elements AR 3.0. Application of the Shapiro–Wilk-test using the statistical software

R 3.2.3 showed normal distribution for carapace length ($p = 0.8688$) and eyestalk width ($p = 0.3054$) among the samples. Subsequently, Welch's Two Sample $t$-test assured that there was no significant difference in measured morphological characters between samples used for cLSM and μCT analysis (carapace length, $p = 0.1925$ ; eyestalk width, $p = 0.1034$). Furthermore, six adult animals of different sizes were kindly provided by Gerhard Scholtz (Humboldt University, Berlin, Germany) and kept as described above. These six specimens were used to trace individual volumetric changes in three stages during sample preparation for μCT. Since the scanning of freshly dissected brains without adding any contrast-enhancing agent turned out to be quite challenging, this approach required the use of larger brains (approximately 170 times larger than those of juveniles). Given that the number of specimens is comparably low as well as we assume that the cytochemical compound of adult brains might differ from that of the juvenile brain (e.g., regarding the content of lipids and proteins), the resulting shrinkage factors have to be interpreted with caution.

## General sample preparation

Genetically identical juveniles of *P. fallax* cf. *virginalis* from the same hatch were taken out of separate tanks with plastic pipettes. They were anaesthetized by chilling them at −18 °C in a beaker with little water for a few minutes. Then, the anaesthetized animals were killed by transferring them into watch glasses containing 4% PFA and phosphate buffered saline (PBS, pH 7.4; 0.1M) for 30 min at room temperature (RT) on the shaker. After that, PFA was removed by immersing the dead specimens at RT on the shaker for two times (5 min each) in fresh PBS for fixation in half-strength Karnovsky's as well as Bouin's solution, or in HEPES-buffered saline (HBS) for fixation in ZnFA before fixation or brain dissection took place. The procedure of anaesthetization and killing of decapod crustaceans is in concordance with the Animal and Welfare Scientific Panel (AHAW) of the *European Food Safety Authority (EFSA) (2005)*.

## Sample preparation for μCT

Before μCT-scanning, seven juvenile siblings were transferred into Eppendorf tubes and immersed in Bouin's solution (10% formaldehyde, 5% glacial acetic acid in saturated aqueous picrinic acid). Fixation took place for one week in a fridge (4 °C). After fixation, the animals were gradually dehydrated in ethanol at RT (30%, 50%, 60%, 70%, 80%, 90%, 96%, and 3× in 99.5% ethanol) for 30 min for each step. For enhancing the contrast, samples were incubated in iodine solution (2% iodine resublimated (cat. #X864.1; Carl Roth GmbH 1 Co. KG, Karlsruhe, Germany) in 99.5% ethanol) for 24 h in the fridge. Iodine was subsequently washed out of the samples for 5 × 3 min with ethanol. Wet specimens were scanned in a glue sealed pipette tip filled with 99.5% ethanol (according to *Sombke et al., 2015*).

## μCT-Scanning

The scans were performed with a laboratory scaled X-ray-microscope (Xradia MicroXCT-200; Carl Zeiss Microscopy GmbH, Jena, Germany) entailing geometric and optical magnification. The Bouin-fixed samples of *P. fallax* cf. *virginalis* were scanned with

a 20× objective in 99.5% ethanol with a voltage of 40 kV, a current of 200 μA, and an exposure time of 8 s. For all scans, binning 2 was applied (summarizing four pixels for noise reduction). Projections obtained by the tomography were reconstructed using the software XMReconstructor (Carl Zeiss Microscopy GmbH, Jena, Germany). To avoid consequent information loss, binning 1 (full resolution) was applied for the following reconstruction resulting in image stacks of 993 × 993 pixels and a pixel size of about 1.1 μm. All original image stacks are openly available (see section "data processing") for scientific reproducibility (*Davies et al., 2017*).

For the inference of an individual in vivo volume during μCT sample preparation and the comparison of chemically-induced effects on the brain volume, brains were dissected under tap water at RT from a total of another six adult animals of different sizes, and each was immediately scanned (stage 1) moistly within a sealed Eppendorf tube at 20 kV and 3 W for 1 s (binning 4) with 400 projections reducing the total scan time to 15 min. Brains were subsequently fixed in Bouin's solution overnight in the fridge. Bouin's solution was replaced two times by tap water and the brain was immediately scanned (stage 2) again moistly using the identic scanning parameters. Afterwards, these fixed brains were gradually dehydrated as outlined above in addition with contrast enhancement by dissolved iodine (2%) in ethanol before being scanned a third time (stage 3) in 99.5% ethanol using again the same scanning parameters.

## Sample preparation for cLSM
### Fixation in half-strength Karnovsky's solution for enhancing autofluorescence of nervous tissue
The two pairs of antennae as well as the eyestalks were removed and the brains of a total of six specimens were dissected in PBS (0.1 M; pH 7.4) using precision forceps (DUMONT®; type 55). The brains were fixed in half-strength Karnovsky's fixative, a mixture of 2% glutaraldehyde (Cat-No. 16220; Electron Microscopy Sciences, Hatfield, PA, USA) and 2% PFA (Cat-no. 0335.2; Carl Roth GmbH 1 Co. KG, Karlsruhe, Germany) in PBS (0.1M; pH 7.4) for a week at 4 °C. Subsequently, a gradual dehydration of the samples was conducted in ethanol (50%, 70%, 80%, 90% for 10 min each, 96% for 30 min, and in 99.5% ethanol for 2 × 30 min) at RT.

### Fixation in zinc-formaldehyde and immunohistochemichal labeling
The whole mount labeling protocol after *Ott (2008)* was applied to improve antibody penetration into nervous tissues, as compared to traditional PFA-fixation. For this method, a total of seven animals were killed by a few drops of formalin instead of 4% PFA in PBS to avoid precipitation of zinc phosphate in combination with ZnFA in the following steps (compare *Ott, 2008*). The central brain was then dissected in HBS as described before and fixed in 4% ZnFA (Cat-No. 15675; Electron Microscopy Sciences, Hatfield, PA, USA) on a shaker for 20 h at RT. The fixed brains were subsequently washed for 3 × 15 min in HBS, and instantly dehydrated and postfixed in Dent's fixative (20% dimethylsulfoxide (DMSO) (Cat-No. 20385; Serva Electrophoresis, Heidelberg, Germany)/80% methanol) in a drop of HBS on the shaker again for 2 h, at RT. Afterwards,

the brains were transferred into 99% methanol. The samples were gradually rehydrated in TRIS-buffer with varying grades of methanol (90%, 70%, 50%, 30% methanol, and finally, pure TRIS-buffer for 15 min each). Samples were preincubated for $2 \times 2$ h in PBS-TX (0.3% triton, 0.02% sodium azide, 1% bovine serum albumin) at RT, followed by incubation in monoclonal mouse anti-synapsin antibody (3C11 anti SYNORF1; Developmental Studies Hybridoma Bank, University of Iowa; deposited by E. Buchner, University Hospital Würzburg, Germany; diluted 1:1 in glycerol) in PBS (1:1,000) for 4 days at 4 °C. Excess primary antibody was washed in PBS-TX for $4 \times 30$ min at RT. Incubation of the Cy 3-conjugated secondary antibody (goat anti-mouse; Jackson Immuno Research, West Grove, PA, USA; 1:1 in glycerol) in HBS (1:500) was carried out in the fridge (4 °C) for 2.5 days. After washing for $2 \times 1$ h in PBS-TX, the samples were dehydrated in ethanolic solutions of different grades (30%, 50%, 70%, 80%, 90%, 96%, and 99.5% for 30 min each) at RT. Regarding antibody specificity, has been shown that the monoclonal mouse-anti-*Drosophila* antibody 3C11 consistently labels synaptic brain regions in representatives of all major subgroups of the malacostracan crustaceans (see *Beltz et al., 2003*; *Harzsch & Hansson, 2008*; *Harzsch, Anger & Dawirs, 1997*; *Harzsch et al., 1999*; *Krieger et al., 2012, 2015*; *Meth, Wittfoth & Harzsch, 2017*; *Vilpoux, Sandeman & Harzsch, 2006*). Hence, it can be assumed that this antibody does in fact label synaptic neuropils in Malacostraca (for more details see *Krieger et al., 2015*).

### cLSM-scanning

Scanning was conducted on a confocal laser-scanning microscope (Leica TCS SP5 II). For optimal light transmission, tissues were cleared in 98% methyl salicylate (Merck, Darmstadt, Germany; Cat-no. W274518). After dehydration in ethanol, brains that were previously fixed in ZnFA ($n = 7$) as well as in half-strength Karnovsky's solution ($n = 6$) were transferred into custom-made scan chambers filled with pure methyl salicylate before confocal laser-scanning. Scanning was performed with an inverted Leica TCS SP5II (Leica, Wetzlar, Germany) using a DPSS-laser with an excitation wavelength of 561 nm and a speed of 400 Hz. For detection of fluorescence (emitted by glutaraldehyde-enhanced autofluorescence as well as by Cy3-conjugates of the secondary antibody), a $10\times$ objective with a numerical aperture of 0.4 was used resulting in stacked images of $1,024 \times 1,024$ pixels with a pixel size of about 0.8 μm. The confocal microscope operated with a pinhole size of 53 μm in diameter and in steps of 1.33 μm (system-optimized to one airy unit and refractive correction for aqueous immersion media).

## Data processing

Volume reconstruction and visualization was carried out using Amira 5.6.0. (FEI Visualization Science Group, Burlington, VT, USA). The central brain, the deutocerebral chemosensory lobes (DCLs) as well as the accessory lobes (AcNs) were segmented manually for volumetric analysis. Three-dimensional surfaces corresponding to the segmentation were generated using unconstrained smoothing (Amira: SurfaceGen). Voxel data of the reconstructed neuropils were extracted by using Amira's material statistics tool. The outline of scans of dissected brains could be instantly visualized without further

| Table 1 Corrected voxelsizes for cLSM-datasets provided at MorphDBase. | | | |
|---|---|---|---|
| Method_fixation | MDB identifier | x/y-voxelsize | z-voxelsize |
| cLSM_Znfa | M64-001.1 | 0.702 | 1.581 |
| cLSM_Znfa | M65-001.1 | 0.8 | 1.581 |
| cLSM_Znfa | M67-001.1 | 0.703 | 1.581 |
| cLSM_Znfa | M68-001.1 | 0.84 | 1.581 |
| cLSM_Znfa | M69-001.1 | 0.876 | 1.581 |
| cLSM_Znfa | M70-001.1 | 0.767 | 1.581 |
| cLSM_Znfa | M66-001.1 | 0.881 | 1.581 |
| cLSM_Glut | M58-001.1 | 0.722 | 1.581 |
| cLSM_Glut | M59-001.1 | 0.868 | 1.581 |
| cLSM_Glut | M60-001.1 | 0.868 | 1.581 |
| cLSM_Glut | M61-001.1 | 0.737 | 1.581 |
| cLSM_Glut | M62-001.1 | 0.682 | 1.581 |
| cLSM_Glut | M63-001.1 | 0.755 | 1.581 |

**Notes:**
All uploaded datasets lost their original (anisotropic) voxelsize due to conversion of the file format. Note that the voxelsizes given for x/y-axis resulted due to slightly varying optical zooming whereas those for the z-axis were modified by using the correction factor for refractive mismatch in methyl salicylate as immersion medium. Use these voxelsizes to display all cLSM-datasets provided at MorphDBase in the correct spatial scaling.

virtual segmentation by using the Amira Isosurface-module. Morphological deformations based on anisometric shrinkages could be detected this way.

Raw data of brain section series (based on μCT as well as cLSM) is available from https://www.morphdbase.de (*Grobe & Vogt, 2009*) under the "media" tab under "Krieger." A combination of the short title "Nischik and Krieger (2017) Marmorkrebs," an identifier according to the specimen, and an abbreviation for the method applied (Nischik and Krieger (2017) Marmorkrebs01_μCT) is given for each of the 20 image stacks in addition to 18 image stacks of brains of another six adult specimens which were scanned (1) freshly dissected prior to fixation; (2) fixed with Bouin's solution overnight but without any dehydration as well as any contrast agent; and (3) after fixation, dehydration in ethanol, and contrast enhancement using 2% iodine. Please note that due to technical requirements of MorphDBase ver. 3.3, all μCT-datasets had to be reduced in color depth from 16 to 8 bit prior to uploading. In contrast to the datasets from μCT-scanning, all cLSM-based datasets feature anisotropic voxels. The original voxel-size is lost due to a conversion of the file format prior to uploading. For displaying these image stacks in the correct spatial relationship, the voxel-size has to be resized for each image stack according to the voxel dimensions given in Table 1 (including the correction factor to minimize refractive mismatch from *Bucher et al. (2000)* of 1.581 in the z-axis).

Furthermore, Table 2 summarizes the volumes of the central brain, of the DCLs, and AcNs of both brain hemispheres according to the three methods applied as well as the z-corrected volumes due to a putative refractive mismatch in cLSM-scans. In Table 3, the volumes of the central brain of six adult specimens are summarized which were analyzed for evaluation of tissue shrinkage throughout the sample preparation (stages 1 to 3) prior to μCT-scanning of fixed and contrast-enhanced samples.

**Table 2 Neuronal volumes in *P. fallax* cf. *virginalis* according to the method applied.**

| Method_fixation | Stack ID | DCL_R (µm³) | DCL_R corrected (µm³) | DCL_L (µm³) | DCL_L corrected (µm³) | AcN_R (µm³) | AcN_R corrected (µm³) | AcN_L (µm³) | AcN_L corrected (µm³) | Central brain (µm³) | Central brain corrected (µm³) | MorphDBase-ID |
|---|---|---|---|---|---|---|---|---|---|---|---|---|
| µCT_Bouin | 1 | 1,486,000 | | 1,576,000 | | 3,646,000 | | 3,667,000 | | 27,713,000 | | J_Krieger_20170807-M-53-001.1 |
| µCT_Bouin | 2 | 1,699,000 | | 1,600,000 | | 4,129,000 | | 4,649,000 | | 32,730,000 | | J_Krieger_20170807-M-51-001.1 |
| µCT_Bouin | 3 | 1,555,000 | | 1,388,000 | | 3,789,000 | | 3,417,000 | | 27,083,000 | | J_Krieger_20170807-M-52-001.1 |
| µCT_Bouin | 4 | 1,274,000 | | 1,274,000 | | 2,942,000 | | 2,896,000 | | 23,533,000 | | J_Krieger_20170807-M-54-001.1 |
| µCT_Bouin | 5 | 1,515,000 | | 1,389,000 | | 4,809,000 | | 3,955,000 | | 29,820,000 | | J_Krieger_20170807-M-55-001.1 |
| µCT_Bouin | 6 | 1,585,000 | | 1,554,000 | | 4,292,000 | | 3,990,000 | | 29,373,000 | | J_Krieger_20170807-M-56-001.1 |
| µCT_Bouin | 7 | 1,171,000 | | 1,171,000 | | 2,646,000 | | 2,696,000 | | 21,071,000 | | J_Krieger_20170807-M-57-001.1 |
| cLSM_Glut | 1 | 791,003 | 948,874 | 883,588 | 1,059,936 | 1,907,300 | 2,287,963 | 2,010,832 | 2,412,159 | 17,744,962 | 21,286,542 | J_Krieger_20170807-M-58-001.1 |
| cLSM_Glut | 2 | 1,041,729 | 1,249,638 | 921,645 | 1,105,587 | 2,635,866 | 3,161,934 | 2,220,232 | 2,663,347 | 19,201,752 | 23,034,055 | J_Krieger_20170807-M-59-001.1 |
| cLSM_Glut | 3 | 984,659 | 1,181,176 | 1,141,834 | 1,369,721 | 2,680,336 | 3,215,275 | 2,665,802 | 3,197,841 | 19,883,410 | 23,851,759 | J_Krieger_20170807-M-60-001.1 |
| cLSM_Glut | 4 | 1,094,224 | 1,312,612 | 1,250,843 | 1,500,489 | 3,309,123 | 3,969,565 | 3,303,236 | 3,962,502 | 27,828,492 | 33,382,528 | J_Krieger_20170807-M-61-001.1 |
| cLSM_Glut | 5 | 1,119,395 | 1,342,806 | 1,091,079 | 1,308,838 | 2,581,602 | 3,096,842 | 2,919,363 | 3,502,015 | 21,709,594 | 26,042,433 | J_Krieger_20170807-M-62-001.1 |
| cLSM_Glut | 6 | 1,084,137 | 1,300,512 | 1,246,876 | 1,495,731 | 2,430,375 | 2,915,435 | 2,719,847 | 3,262,680 | 20,655,140 | 24,777,541 | J_Krieger_20170807-M-63-001.1 |
| cLSM_ZnFA | 1 | 817,084 | 980,155 | 619,376 | 742,989 | 2,226,071 | 2,670,344 | 1,165,159 | 1,397,698 | 18,915,004 | 22,690,055 | J_Krieger_20170807-M-64-001.1 |
| cLSM_ZnFA | 2 | 1,343,450 | 1,611,573 | 1,111,938 | 1,333,857 | 2,760,117 | 3,310,976 | 2,425,304 | 2,909,342 | 25,255,030 | 30,295,377 | J_Krieger_20170807-M-65-001.1 |
| cLSM_ZnFA | 3 | 1,419,327 | 1,070,987 | 892,802 | 1,702,596 | 3,115,787 | 3,172,392 | 2,644,587 | 3,737,634 | 25,096,024 | 30,104,705 | J_Krieger_20170807-M-66-001.1 |
| cLSM_ZnFA | 4 | 1,198,716 | 1,437,957 | 1,161,789 | 1,393,660 | 2,756,689 | 3,306,871 | 2,440,600 | 2,927,697 | 23,043,944 | 27,643,055 | J_Krieger_20170807-M-67-001.1 |
| cLSM_ZnFA | 5 | 1,411,353 | 1,693,029 | 1,474,522 | 1,768,805 | 3,355,852 | 4,025,608 | 3,589,474 | 4,305,856 | 27,222,322 | 32,655,320 | J_Krieger_20170807-M-68-001.1 |
| cLSM_ZnFA | 7 | 1,465,908 | 1,758,472 | 1,376,278 | 1,650,954 | 3,515,581 | 4,217,219 | 3,112,145 | 3,733,264 | 25,117,220 | 30,130,071 | J_Krieger_20170807-M-69-001.1 |
| cLSM_ZnFA | 8 | 815,546 | 1,086,852 | 906,027 | 978,312 | 2,123,881 | 2,651,125 | 2,210,045 | 2,547,765 | 15,918,198 | 19,095,127 | J_Krieger_20170807-M-70-001.1 |

**Notes:**
Note that corrected volumes use a *z*-axial correction factor of 1.581 to minimize refractive mismatch due to scanning in methyl salicylate.
DCL, deutocerebral chemosensory lobe; AcN, accessory neuropil (lobe).

**Table 3 Brain volumes of six adult specimens of *P. fallax* cf. *virginalis* during three stages of sample preparation for μCT-scanning.**

| Method_ fixation | Specimen ID | Central brain ($\mu m^3$) | MorphDBase-ID |
|---|---|---|---|
| μCT_stage_1 | 1 | 3,390,200,000 | J_Krieger_20180329-M-71-001.1 |
| μCT_stage_2 | 1 | 3,044,452,000 | J_Krieger_20180329-M-77-001.1 |
| μCT_stage_3 | 1 | 3,224,483,000 | J_Krieger_20180329-M-83-001.1 |
| μCT_stage_1 | 2 | 6,524,711,000 | J_Krieger_20180329-M-72-001.1 |
| μCT_stage_2 | 2 | 6,010,940,000 | J_Krieger_20180329-M-78-001.1 |
| μCT_stage_3 | 2 | 6,409,148,000 | J_Krieger_20180329-M-84-001.1 |
| μCT_stage_1 | 3 | 6,548,093,000 | J_Krieger_20180329-M-73-001.1 |
| μCT_stage_2 | 3 | 5,544,624,000 | J_Krieger_20180329-M-79-001.1 |
| μCT_stage_3 | 3 | 5,022,508,000 | J_Krieger_20180329-M-85-001.1 |
| μCT_stage_1 | 4 | 7,620,066,000 | J_Krieger_20180329-M-74-001.1 |
| μCT_stage_2 | 4 | 4,925,074,000 | J_Krieger_20180329-M-80-001.1 |
| μCT_stage_3 | 4 | 5,451,010,000 | J_Krieger_20180329-M-86-001.1 |
| μCT_stage_1 | 5 | 6,041,379,000 | J_Krieger_20180329-M-75-001.1 |
| μCT_stage_2 | 5 | 3,333,409,000 | J_Krieger_20180329-M-81-001.1 |
| μCT_stage_3 | 5 | 3,098,399,000 | J_Krieger_20180329-M-87-001.1 |
| μCT_stage_1 | 6 | 7,333,405,000 | J_Krieger_20180329-M-76-001.1 |
| μCT_stage_2 | 6 | 4,984,792,000 | J_Krieger_20180329-M-82-001.1 |
| μCT_stage_3 | 6 | 5,115,247,000 | J_Krieger_20180329-M-88-001.1 |

Notes:
Stage (1) freshly dissected brain scanned moistly in water; stage (2) after fixation in Bouin's solution but without dehydration or contrast enhancement; stage (3) after fixation, dehydration in ethanol and contrast enhancement in 2% iodine.

## Statistics

The acquired volume data (Fig. 1) were exported into Microsoft® Excel for descriptive statistics. Statistical analyses were performed with the "stats"-package and illustrated with the "graphics"-package of R 3.2.3. A one-way analysis of variance (ANOVA) was carried out for calculation of statistical significant differences between volumetric data and the methods used (Fig. 2). The effects of treatment on the volume of the subunits DCLs and AcNs, as well as on the total volume of the central brain were calculated for each structure by Tukey's post hoc test. A paired two sample Student's *t*-test was applied for volumes of left and right lobe of AcN and DCL respectively, to test reproducibility of the manual volume reconstruction, which was conducted by the same investigator for each specimen. A Wilcoxon signed rank test was applied to test the effect of chemically induced difference in the brain volume throughout the sample preparation for μCT-scanning.

## RESULTS

### Micro-computed X-ray microscopy

Tomograms were generated with μCT to morphometrically analyze the brain of *P. fallax* cf. *virginalis*. Volume rendering allowed virtually sectioning of the animal in different planes for neuroanatomical analysis as well as visualizing the brain in its natural position within the cephalon (Figs. 3A–3D). The image contrast as a function of the tissue density

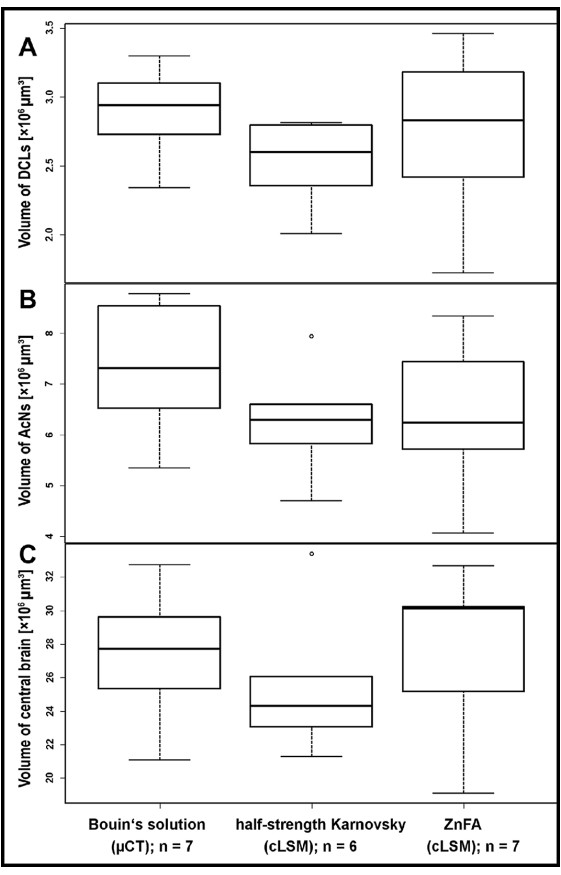

**Figure 1 Volumes of brain compartments according to the methods applied.** Boxplots with median, upper, and lower quartiles, minima and maxima, and outliers of volumes of the deutocerebral che­mosensory lobes (DCLs) in (A), the accessory lobe (AcNs) in (B), and the central brain in (C). Note that the displayed volumes for DCLs and AcNs refer to the total volumes of both brain hemispheres. All volumetric data of cLSM-scanning resulted from *z*-corrected datasets using the correction factor for refractive mismatch with methyl salicylate provided by *Bucher et al. (2000)*.

was high enough to allow identification of single neuropils (compare Figs. 4A and 4B). Thus, the central brain including the DCLs and the AcNs could be reconstructed (Figs. 3C and 3D). The central brain is vertically curved and held in its upright position by the protocerebral tract originating in the lateral protocerebrum (latPC; Figs. 3C and 3D) within the eyestalks as well as by the nerves of the antennae 1 and 2 (A$_I$ and A$_{II}$; Figs. 3A and 3C). Here, an approximate in situ coherent reconstruction of the neuropils in their spatial context is assumed, since only a few steps of sample preparation were required for scanning an animal as a whole.

While μCT-scanning of contrasted tissues delivered a high image contrast, the analysis of μCT-scans of unfixed as well as fixed brains (without any contrast agent) of another six adult specimens, which were used for inferring individual in vivo volumes during μCT sample preparation, turned out to be fairly challenging. In freshly dissected as well as in Bouin-fixed but uncontrasted brains, the image contrast was barely high enough to identify the limits of the tissue surfaces but did not suffice to identify

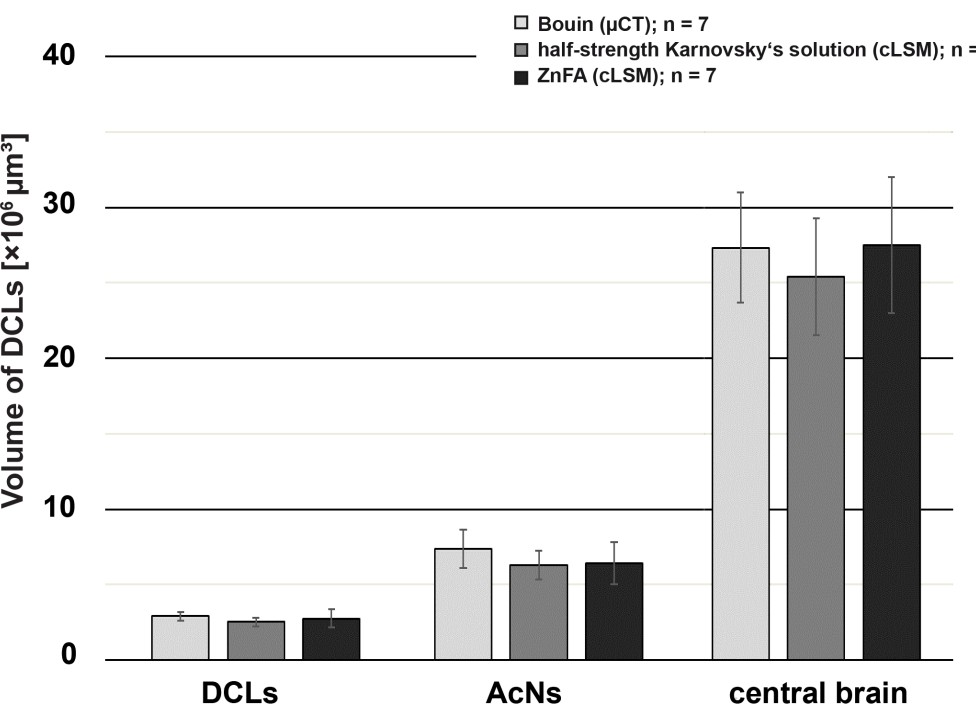

**Figure 2 Comparison of volumetric data based on the methodologic approach of *P. fallax* cf. *virginalis*.** Each bar represents the average of reconstructed volumes of deutocerebral chemosensory lobes (DCLs), accessory lobes (ACNs), as well as of the central brain of all specimens analyzed. Note that volumes of the DCLs as well as of the AcNs are plotted pairwise (both brain hemispheres per specimen). Applied methods: µCT (Bouin-fixation), cLSM (half-strength Karnovsky's solution), and cLSM (ZnFA-fixation). The levels of significance between pairs of volumes of brain substructures according to the methods applied are based on Tukey's post hoc test from a one-way analysis of variance (ANOVA). Note that all cLSM volumes include the correction factor for refractive mismatch for methyl salicylate from *Bucher et al. (2000)*.

internal substructures such as neuropils or cell clusters. Therefore, the use of larger brains of adult specimens (approximately 170 times larger than those of juveniles) was essential. The central brain volume resulted in a significant average shrinkage of 24% (ranging from 7.9% to 44.8%) after fixation in Bouin's solution (Wilcoxon signed rank test: $p = 0.0313$; $n = 6$) and no further significant shrinkage (Wilcoxon signed rank test: $p = 0.6875$; $n = 6$) after dehydration and contrast-enhancement using 2% iodine in ethanol (Fig. 5; Table 3).

## Confocal laser-scanning microscopy
### *Autofluorescence in half-strength Karnovsky's solution*
Tissues fixed with half-strength Karnovsky's solution resulted in cLSM scans characterized by a high resolution and remarkable signal to noise ratio. Furthermore, intricate histological

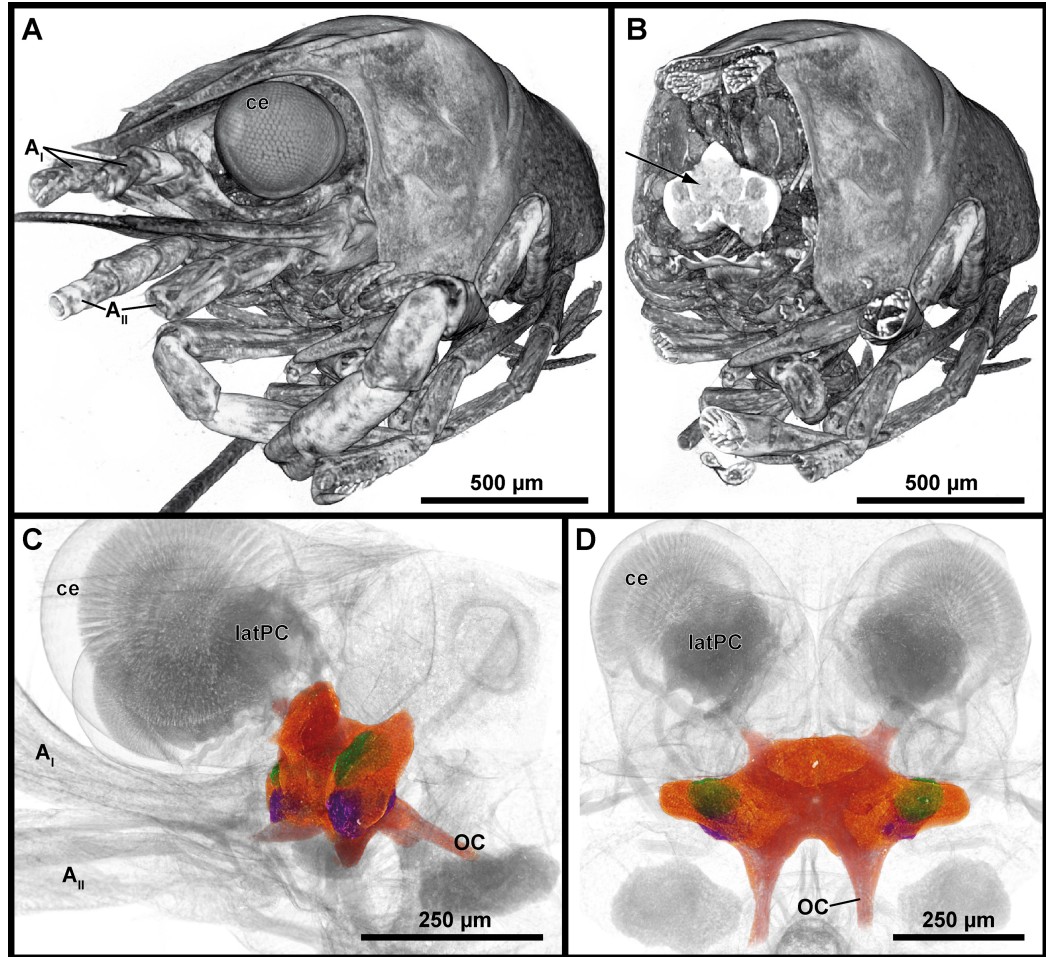

**Figure 3 Volume renderings of the brain in *P. fallax* cf. *virginalis* from a μCT-scan.** The outline of the whole animal body is visualized in (A), while in (B), a virtual cutaway reveals the position of the brain (white area indicated by a black arrow) from the same perspective. The color-labeled areas in (C) and (D) are based on surface reconstruction and show the central brain (orange), the deutocerebral chemosensory lobes (DCLs: green), and the accessory lobes (AcNs: purple). Note that the lateral protocerebrum (latPC) is not reconstructed but visible in dark gray in (C) and (D) beneath the ommatidia of the complex-eyes (ce). Study sites: AI, antenna 1 (antennule); AII, antenna 2 (antenna); OC, oesophageal connective.

details such as individual olfactory glomeruli in the DCL and even microglomeruli within the AcN were resolved with high accuracy (compare Figs. 4A and 4C).

### *Immunohistochemistry, ZnFA-fixation*

In comparison to the previously described method, volume divergences may only be the result of the fixation of the specimen—in this case using the ZnFA-fixation protocol after *Ott (2008)* and, of course, of the individual bias, due to manual segmentation for three-dimensional reconstruction in Amira. In fact, the reconstructed volumes differ only slightly, and thus, no statistically significant difference between the applied fixations (ZnFA and half-strength Karnovsky's solution) could be detected for each of the brain substructures analyzed (Fig. 2). However, a slightly higher average volume indicates,

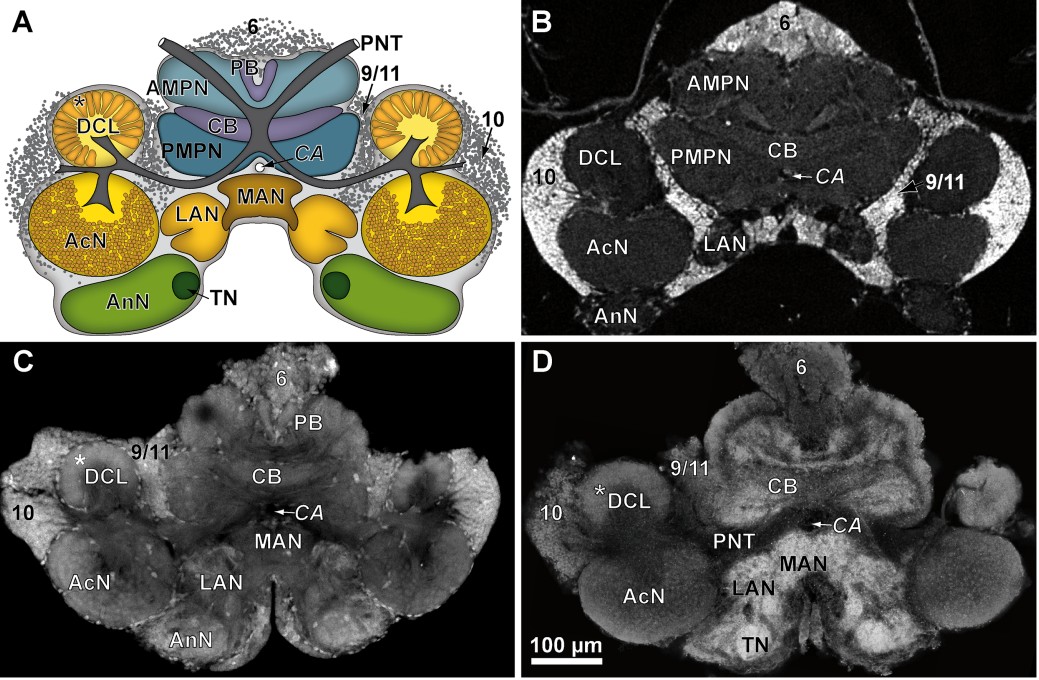

**Figure 4 Comparison of virtual brain sections of *P. fallax* cf. *virginalis* based on all three methods applied.** The scheme in (A) illustrates the general organization of the central brain (omitting the lateral protocerebrum, nerves, and the oesophageal connectives). Frontal virtual sections of the central brain, according to the tomographic and corresponding fixing method applied, are shown equally scaled in (B) (µCT using Bouin's solution); in (C) (cLSM using autofluorescence of half-strength Karnovsky's solution); and in (D) cLSM using ZnFA and immunohistochemical labeling against synapsin. Study sites: AcN, accessory neuropil; AMPN, anterior medial protocerebral neuropil; AnN, Antenna-II-neuropil; CA, cerebral artery; CB, central body; DCL, deutocerebral chemosensory lobe (olfactory lobe); LAN, lateral antenna-I-neuropil; MAN, median antenna-I-neuropil; PB, protocerebral bridge; PMPN, posterior medial protocerebral neuropil; PNT, projection neuron tract; TN, tegumentary neuropil; 6, 9/11, and 10 indicate somata clusters (6), (9/11), and (10); asterisk indicates olfactory glomeruli in (A), (C), and (D).

that the whole brain is somewhat better preserved by using the fixation with ZnFA and dehydrogenation in methanol and DMSO. Since an immunohistochemical labeling was used here, the resulting tissue contrast is different to that of autofluorescence in half-strength Karnovsky's solution (compare Figs. 4C and 4D). In this case, primarily areas in which the synaptic membrane protein synapsin was present were stained (Fig. 4D). While the intention was to facilitate the identification of neuropil regions, it actually increased image noise.

### Micro-computed X-ray microscopy vs confocal laser-scanning microscopy

All three methods very likely differ to a certain degree from in vivo congruent nervous tissue volume. Since all samples at least temporarily were exposed to hyperosmolar media resulting in dehydration, it has to be assumed that artifacts due to shrinkage most likely occurred. An expansion of the tissue seems unlikely, since also in the wet-scan µCT approach, dehydration takes place (e.g., fixation in Bouin's solution, dehydration in ethanol). However, neither the general imaging technique (µCT *vs* cLSM: Welch's

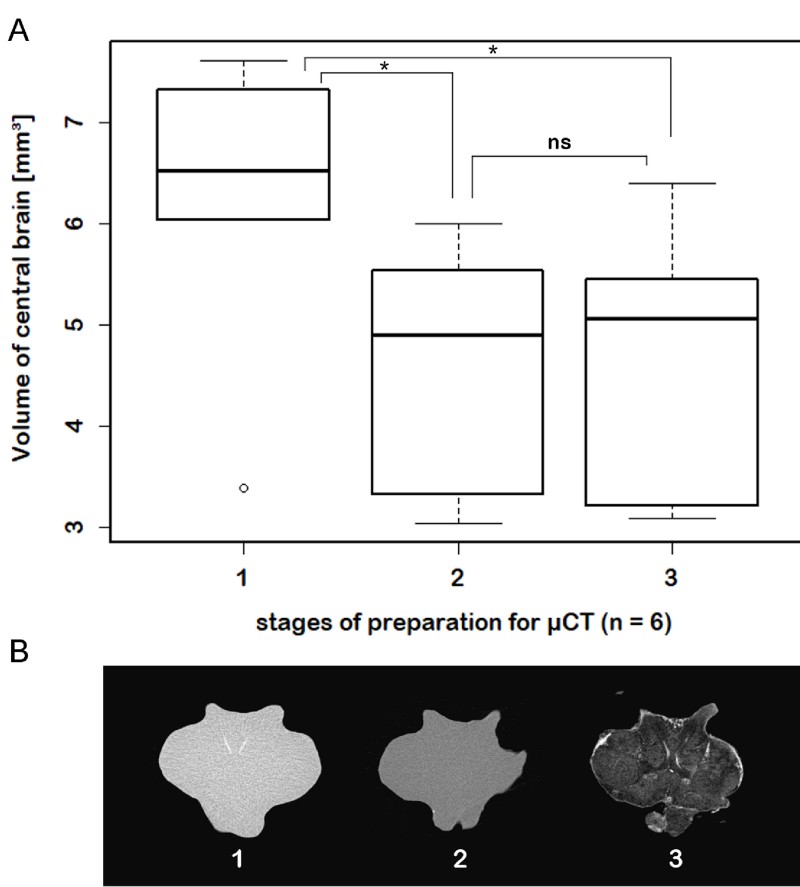

**Figure 5 Virtual sections of adult individual of *P. fallax* cf. *virginalis* and volumes of central brains after different preparation stages.** (A) Boxplots with median, upper, and lower quartiles, minima and maxima, and outliers of volumes of the central brain after different stages of preparation for μCT wet-scanning. (B) Frontal virtual slices of the central brain of an adult individual of *P. fallax* cf. *virginalis*. Stage (1) The brains are just dissected and immediately scanned in tap water. Stage (2) Brains are fixed in Bouin's solution. Stage (3) Gradual dehydration and contrast enhancement with iodine (2%). \*: Significant difference between stage 1 and 2, 3 (Wilcoxon signed rank test: $p = 0.0313$; $n = 6$). ns: No significant difference between stage 2 and 3 (Wilcoxon signed rank test: $p = 0.6875$; $n = 6$).

Two Sample *t*-test: $p = 0.2136$; $n = 20$), nor the choice of fixative between both cLSM approaches did influence the tissue volumes significantly (Welch's Two Sample *t*-test: $p = 0.4541$; $n = 13$). Assuming a symmetrical development of both brain hemispheres, the investigator's individual bias in manual segmentation for three-dimensional reconstruction was tested by comparing volumes of both hemispheres. While the volume of the AcNs reliably exceeded the volume of the DCLs (single-factor ANOVA: $p < 0.001$; $n = 20$), no difference in the volume of the left and right hemispheres of paired lobes could be found in any given treatment (paired Student's *t*-test: DCL: $p = 0.2691$; AcN: $p = 0.1107$; $n = 20$).

The highest DCLs-volumes (total volume of both hemispheres) were reconstructed in specimens which were fixed in Bouin's solution and imaged by the use of μCT (mean DCLs volume [mm³]: $0.0029 \pm 0.0003$; $n = 7$). The DCLs volumes in specimens fixed in

ZnFA (mean DCLs volume [mm$^3$]: $0.0027 \pm 0.0006$; $n = 7$) were insignificantly higher compared to specimens fixed in half-strength Karnovsky's solution (mean DCLs volume [mm$^3$]: $0.0025 \pm 0.0003$; $n = 6$). The measured volume of nervous tissue was most congruent to assumed in vivo volume when specimens were fixed in Bouin and scanned in ethanol with µCT (Figs. 1, 2, and 5). This is followed by cLSM visualization of whole-mounts fixed in ZnFA and at last of tissues fixed in half-strength Karnovsky's solution (compare Figs. 1A, 1B, and 1C). Although µCT derived volumes are significantly smaller than those of freshly dissected brains (average of 24%, ranging from 7.9% to 44.8%), cLSM tomograms of tissues fixed in half-strength Karnovsky's solution reveal a further relative shrinkage of 12.5%; and of 5% when using ZnFA-fixation.

In *P. fallax* cf. *virginalis*, light-microscopic imaging techniques on the brain require the removal of the cuticle or dissection of the nervous tissue, thus an analysis of the brain within the animal is only feasible by using X-ray -or magnetic resonance imaging techniques (*Brinkley et al., 2005*; *Herberholz et al., 2004*; *Köhnk et al., 2017*, and reviewed in *Ziegler et al., 2011*). Three-dimensional surfaces of half-strength Karnovsky's solution-fixed brains, based on isosurfaces using a grayscale-threshold (Amira: Isosurface), appeared more porous and wrinkled (compare Figs. 6A–6C), while the ZnFA-fixed brains had a smoother appearance (compare Figs. 6D–6F). Along the neuraxis, brains appear more furrowed after fixation with half-strength Karnovsky's solution compared to those fixed with ZnFA (Fig. 6). In addition, the bilaterally symmetric antenna-2-nerve (A$_{II}$Nv; Figs. 6D and 6F) was better preserved in its fibrous organization in ZnFA preparations compared to those fixed with half-strength Karnovsky's solution (Figs. 6A and 6C).

For comparison of image quality, a schematic overview of the central brain (Fig. 4A) as well as virtual frontal sections recorded on a similar anatomical plane (approximately as shown in Fig. 3B) are shown in Fig. 4B–4D. In images generated by cLSM within the DCL, individual olfactory glomeruli are visible (Figs. 4C and 4D). The images produced by µCT have a lower resolution and a lower signal-to-noise ratio. Here, the identification of individual olfactory glomeruli at least at the chosen magnification is not possible (Fig. 4B). In fact, the highest contrast was obtained by confocal imaging of half-strength Karnovsky's solution-fixed tissue (Fig. 4C). By the use of autofluorescent enhancement of half-strength Karnovsky's solution-fixation, the sample preparation is much less time consuming but more unspecific than applying the two-step protocol of immunohistochemichal labeling. The high resolution in both methods reveals details like individual olfactory glomeruli of the DCLs as well as even microglomeruli within the AcNs (Figs. 4A, 4C, and 4D).

## DISCUSSION

Confocal laser-scanning microscopy turned out to be the most suitable technique for identification of neuropils and neuronal somata (Figs. 4A, 4C, and 4D). Nevertheless, for the use of brain morphometry at a coarse level, the image contrast is sufficient to distinguish the neuropils from the surrounding tissue in all applied techniques. In contrast to the preparation for µCT, the central brain has to be dissected for cLSM whole mounts, which comes with some disadvantages. The process of sample preparation

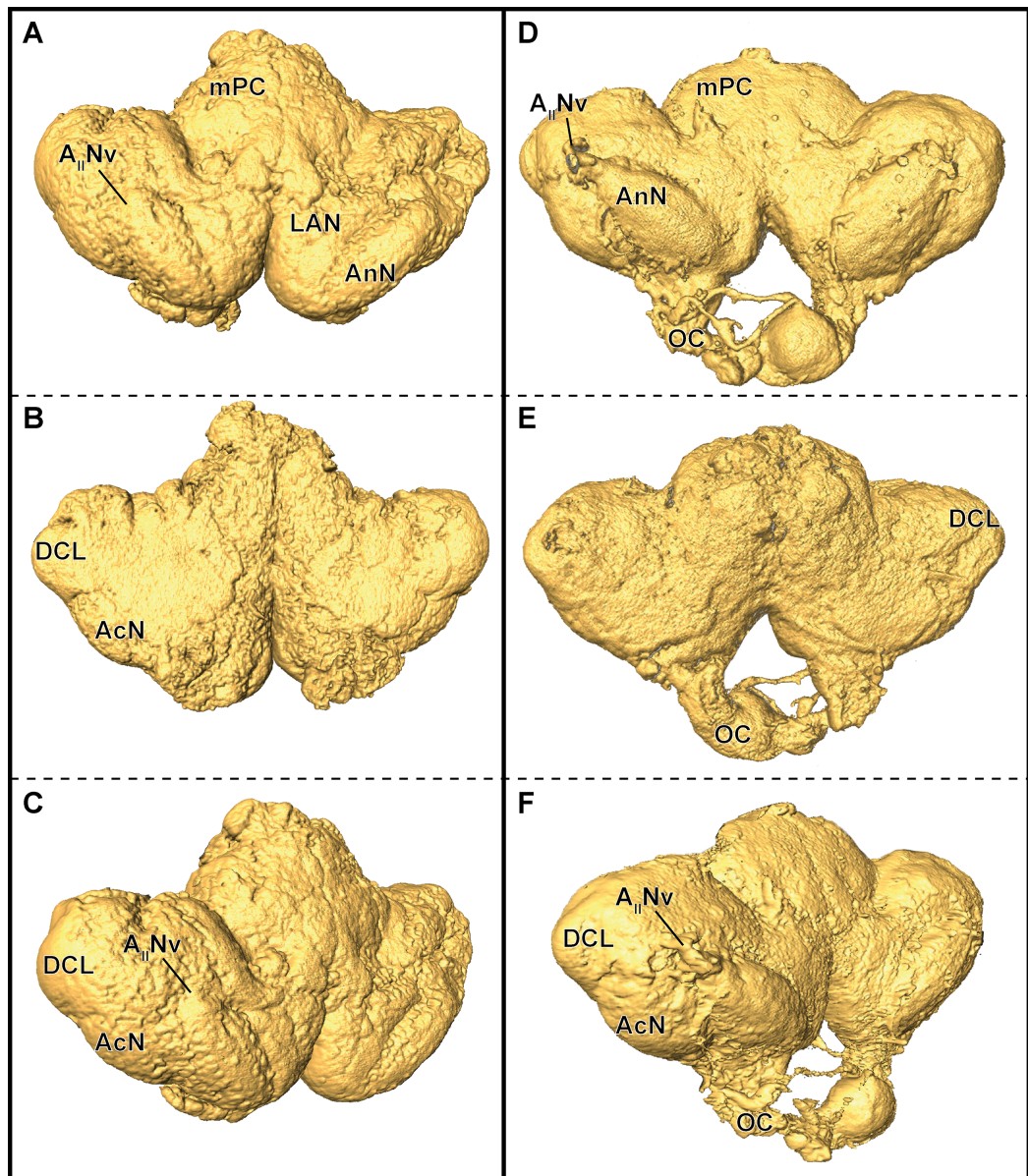

**Figure 6 Comparison of grayscale-based surface reconstructions of the central brain of two individuals of *P. fallax* cf. *virginalis*.** The isosurfaces are based on confocal laser-scanning microscopy of dissected brains fixed in half-strength Karnovsky's solution (A–C) and in ZnFA (D–F), and are shown from different perspectives (A and D: from anterioventral; B and E: from posteriodorsal; C and F: from ventrolateral). Study sites: AIINv, antenna 2-nerve; AcN, accessory neuropil; AnN, antenna 2-neuropil; DCL, deutocerebral chemosensory lobe (olfactory lobe); LAN, lateral antenna 1-neuropil; mPC, median protocerebrum; OC, oesophageal connective.

alters the brain's characteristic shape, which in situ is displaying a curvature along the neuraxis. Due to dissection, the originally upstanding central brain collapses so that spatial relationships of brain regions become artificial. In addition, the use of methyl salicylate for clearing in advance of cLSM-scanning has been reported to cause unpredictable shrinkage of nervous tissues ranging from 3.5% to 27% of the volume (*Bucher et al., 2000*). Furthermore, a refractive mismatch along the *z*-axis must be

considered for the use of methyl salicylate as mounting medium for cLSM which leads to enormous aberrations of volume (*Bucher et al., 2000*). Using the *z*-axial correction factor to eliminate refractive mismatch for the use of methyl salicylate provided by *Bucher et al. (2000)* of 1.581 (for a 10× dry objective with a numerical aperture of 0.40) resulted in spatial congruence of the brain dimensions that almost equals those based on μCT-scans. The resulting volumes of brains fixed in ZnFA as well as in half-strength Karnovsky's solution were still insignificantly smaller (ZnFA *vs.* Bouin: 5%; $p = 0.826$; half-strength Karnovsky's solution *vs.* Bouin: 12.5%; $p = 0.362$; one-way ANOVA) than those obtained from μCT-scans. The refractive mismatch is often altered by the anisometric distribution of clearing agent, different refractive indices of the tissues analyzed and thus, its progression along the *z*-axis is not linear (*Hell et al., 1993*; *Besseling, Jose & Blaaderen, 2015*). The anisometry of cLSM-stacks as a result of optical limitations and the associated necessity of a *z*-axial correction factor prior to analysis hence adds another source of inaccuracy compared to μCT-datasets. Although, brain volumes based on cLSM-stacks that were axially rescaled are insignificantly smaller, the sample preparation seems to have a higher influence on tissue shrinkage than the sample preparation for μCT-scanning. Also, the accuracy of volumetric analysis is decreased when using the fixation of ZnFA in addition with immunohistochemical labeling against synapsins compared to the sample preparations using Bouin and half-strength Karnovsky's solution, resulting in higher variances. One explanation could be that since the latter two approaches (using Bouin as well as half-strength Karnofsky's solution) are based on the detection of differences in tissue densities, the sample preparation using ZnFA is basically dependent on the cross-linking of the antibody against SYNORF1, and thus the distribution of the target epitopes.

While volumetry can be helpful to analyze differences in the size relation of brain structures within an organism as well as for interspecific comparison, referring brain size as a function of cognitive capacity is highly debated (*Chittka & Niven, 2009*; *Healy & Rowe, 2013*). However, for example in honeybees (*Durst, Eichmüller & Menzel, 1994*; *Groh, Ahrens & Rössler, 2006*) or in leaf-cutting ants (*Groh et al., 2014*), morphometric analysis showed an age and experience dependent difference in neuropil volume of the mushroom body. Especially for the volumetry of these tiny brain subcompartments, measurement errors can have a huge impact and should therefore be minimized. It is therefore important to consider the osmolality of immersive chemicals (fixatives and buffers) in respect to the target tissue. Increased tissue shrinkage was reported with increasing hypertonia (*Rasmussen, 1974*), hence chemical fixatives with similar osmolality as the original osmotic environment (e.g., <280 mOsm/kg $H_2O$ for freshwater species) are recommended (*Bullock, 1984*; *Coetzee & van der Merwe, 1985*). An area shrinkage of 11–20% was reported for rabbit corneal endothelial cells fixed with Karnovsky fixative (850 mOsM/kg; *Doughty, Bergmanson & Blocker, 1997*). However, in the current study merely a half-strength Karnovsky fixative (640 mOsM/kg; *Platt, Oliver & Thomson, 1997*) was applied. *Ott (2008)* described a new fixation protocol for immunohistochemical staining, which uses ZnFA instead of PFA resulting in lower osmolality (325 mOsm/kg). This has been shown to improve antibody penetration and preservation of spatial brain morphology. *Ott (2008)* demonstrated effects of different fixation protocols on
morphological preservation on the brain of the desert locust *Schistocerca gregaria*. Ott showed that fixation using PFA lead to increased wrinkling in contrast to ZnFA-fixation. Likewise, the nature of dehydration and the duration of fixation were found to be decisive factors. This corresponds to findings from *Ott (2008)*, as preparations with a shorter fixing time in ZnFA showed a lower wrinkling. Here, the cLSM-tomograms resulted in smoother surfaces of brains analyzed.

Apart from indirect volumetric measurements evaluated here, a possible approach to obtain approximate in vivo volumes for invertebrate brains, would be the use of a micro-volumeter according to *Douglass & Wcislo (2010)*, as an example for direct volumetry. By the use of Archimedes' principle, the dissected brain is put in a liquid-filled tube and the occurring volume displacement of the liquid can be measured with a micro-pipette stepwise until the previous meniscus is reached. In this way, the total removed volume equals the actual brain volume. However, based on initial trials, instrumental errors still outweigh a precise measurement of very small volumes (<1 ml). This technique needs more precise adjustment for readout of the meniscus. A successful application of direct volumetry using Archimede's principle or high resolution magnetic resonance imaging of living animals will offer a conclusive reference for in vivo brain volumes. Although the sample size is not markedly high ($n$ = ranging from six to seven per treatment), we consider that specimens prepared for μCT feature the closest in vivo coherence (24% average shrinkage; ranging from 7.9% to 44.8%). We could show that shrinkage due to each single step throughout the sample preparation was primarily influenced by the fixation in Bouin, whereas the subsequent steps of this preparation interestingly did not contribute substantially to tissue shrinkage. However, dehydration in ethanol in addition with contrast-enhancement by iodine (*Buytaert et al., 2014*) as well as the scanning procedure itself (*Gianoncelli et al., 2015*) indeed most likely lead to a deviation from the in vivo volume.

The deutocerebral chemosensory and AcNs are convenient landmarks for a volumetric evaluation of experimental effects, due to their conspicuous structure with an almost spherical shape, they can be easily identified. Especially for the conspicuous DCLs, volumetric data are available for a considerable number of crustacean species (see *Beltz et al., 2003*; *Krieger et al., 2015*; *Tuchina et al., 2015*; and review in *Schmidt, 2016*). In particular, the relative volumes of these homologous brain regions vary greatly among species. For comparison, Table 4 displays the volume information for the DCLs of selected crustacean species accompanied by the data obtained for *P. fallax* cf. *virginalis*. However, it should be noted that all individuals of *P. fallax* cf. *virginalis* analyzed for the comparison of imaging techniques were juvenile siblings with a body length of about 5 mm. Adults of this species can reach a body length of 120 mm, and thus feature markedly larger brain sizes (ranging up to 300 times higher volumes than those of the juveniles analyzed here). Furthermore, it can be expected that methods including chemicals do not influence volumes of different brain substructures alike (such as somata or fibers). Since the main volume of the brain is dominated by neuropils as well as the fact that all three methodological approaches did not meet the

**Table 4 Average volume of deutocerebral chemosensory lobe (single lobe) of various decapod species.**

| Taxon | Species | n | DCL volume [mm³] *with optical correction | Reference |
|-------|---------|---|-------------------------------------------|-----------|
| Astacida | *Procambarus fallax* cf. *virginalis* (juv.) | 7 | 0.0015 | This study (μCT) |
| Astacida | *Procambarus fallax* cf. *virginalis* (juv.) | 7 | 0.0011/*0.0014 | This study (ZnFA) |
| Astacida | *Procambarus fallax* cf. *virginalis* (juv.) | 6 | 0.0011/*0.0013 | This study (Karnofsky) |
| Astacida | *Procambarus clarkii* (adult) | 3 | 0.0097 | *Beltz et al. (2003)* |
| Homarida | *Homarus americanus* (adult) | 2 | 0.1412 | *Beltz et al. (2003)* |
| Anomala | *Birgus latro* (adult) | 1 | 0.3747 | *Krieger et al. (2012)* |
| Brachyura | *Sesarma* sp. (adult) | 3 | 0.0061 | *Beltz et al. (2003)* |

Note:
* indicate resulting volumes using the correction factor for methyl salicylate provided as immersion medium by *Bucher et al. (2000)*.

requirements of a differentiated volumetric analysis, an inhomogeneous tissue shrinkage was neglected for simplicity.

For interspecific comparisons and the corresponding phylogenetic or neuroanatomical value of volumetric data, several factors have to be standardized, such as the experimental procedure including fixatives and technique of volumetry (e.g., histological sections with volumetric extrapolations based on the section thickness, or 3D-reconstruction of tomographic data). Shrinkage factors, based on the morphometric method and fixatives used, can serve as a tool to approximate already referred volumetric data to a common denominator for a more reasonable interspecific comparison. Because of its parthenogenetic nature, *P. fallax* cf. *virginalis* is well suited generating such a reference system at least for aquatic arthropods featuring a comparable osmolality. Although the juvenile specimens analyzed were genetically identical and of the same age (clutch), individual brain sizes might vary due to phenotypic plasticity as has been shown e.g., for the marmoration pattern in siblings of the Marmorkrebs (review: *Vogt, 2011*), but also for brain sizes in clones of *Daphnia magna* (*Macagno, Lopresti & Levinthal, 1973*), and also in vertebrates (review: *Mitchell, 2007*). Consequently, comparative brain allometry in juvenile isogenetic siblings appears to be the most favorable approximation to neglect inter-individual variations.

## CONCLUSIONS

Although today, neuroanatomical volumetric data are available for a variety of crustacean species, interspecific comparisons often suffer from methodological differences in volumetry. Variations in tissue volume as artefacts of experimental sample preparation, such as fixation procedures, might be incorrectly assigned to biological phenomena. While comparative brain morphometry and especially volumetry as a measure for cognitive capabilities is controversially discussed (*Chittka & Niven, 2009*; *Healy & Rowe, 2013*), it is, however, a useful tool for other fields of interest, such as the neural

development (*Helluy, Ruchhoeft & Beltz, 1995*), neurophylogeny as well as specific evolutionary adaptations of the nervous system (*Beltz et al., 2003*; *Krieger et al., 2015*; reviewed in *Schmidt, 2016*). Therefore, a standardization of method-based deviations is highly recommended. Here, we aimed at a consistent methodological approach to evaluate standard imaging techniques as well as to obtain conversion factors to deduce approximate in vivo volumes based on the method of analysis. Well-founded conversion factors will allow for a posteriori standardization of determined nervous tissue volumes in malacostracans, and therefore help to eliminate the aforementioned sources of error. Due to its parthenogenetic reproduction, *P. fallax* cf. *virginalis* produces genetically identical offspring making it an ideal model organism especially for methodological studies. Further comparative studies covering all standard techniques in the same manner, will offer a conclusive reference system and comparability, irrespective of fixation protocol chosen which is indeed dependent of the imaging technique and the specific scientific question.

## ACKNOWLEDGEMENTS

We cordially thank Gerhard Scholtz (Humboldt-University of Berlin) for the kind provision of animals and Caroline Viertel (University of Greifswald) for the animal husbandry in the laboratory. We like to express our gratitude to Steffen Harzsch and Marie K. Hörnig (University of Greifswald) for reading and improving the manuscript. We would like to thank both anonymous reviewers for their input, as these recommendations led us to markedly improve the manuscript.

### Funding

This work was supported by the German Science Foundation (DFG INST 292/119-1 FUGG and DFG INST 292/120-1 FUGG). The funders had no role in study design, data collection and analysis, decision to publish, or preparation of the manuscript.

### Grant Disclosures

The following grant information was disclosed by the authors:
German Science Foundation: DFG INST 292/119-1 FUGG and DFG INST 292/120-1 FUGG.

### Competing Interests

The authors declare that they have no competing interests.

### Author Contributions

- Emanuel S. Nischik performed the experiments, analyzed the data, prepared figures and/or tables, authored or reviewed drafts of the paper, approved the final draft, performed the statistical analysis.
- Jakob Krieger conceived and designed the experiments, performed the experiments, analyzed the data, prepared figures and/or tables, authored or reviewed drafts of the paper, approved the final draft.

## Data Availability

Raw data of brain section series (based on μCT as well as cLSM) is available from https://www.morphdbase.de:

www.morphdbase.de/?J_Krieger_20180329-M-71.1,
www.morphdbase.de/?J_Krieger_20180329-M-72.1,
www.morphdbase.de/?J_Krieger_20180329-M-73.1,
www.morphdbase.de/?J_Krieger_20180329-M-74.1,
www.morphdbase.de/?J_Krieger_20180329-M-75.1,
www.morphdbase.de/?J_Krieger_20180329-M-76.1,
www.morphdbase.de/?J_Krieger_20180329-M-77.1,
www.morphdbase.de/?J_Krieger_20180329-M-78.1,
www.morphdbase.de/?J_Krieger_20180329-M-79.1,
www.morphdbase.de/?J_Krieger_20180329-M-80.1,
www.morphdbase.de/?J_Krieger_20180329-M-81.1,
www.morphdbase.de/?J_Krieger_20180329-M-82.1,
www.morphdbase.de/?J_Krieger_20180329-M-83.1,
www.morphdbase.de/?J_Krieger_20180329-M-84.1,
www.morphdbase.de/?J_Krieger_20180329-M-85.1,
www.morphdbase.de/?J_Krieger_20180329-M-86.1,
www.morphdbase.de/?J_Krieger_20180329-M-87.1,
www.morphdbase.de/?J_Krieger_20180329-M-88.1,
www.morphdbase.de/?J_Krieger_20170807-M-70.1,
www.morphdbase.de/?J_Krieger_20170807-M-69.1,
www.morphdbase.de/?J_Krieger_20170807-M-68.1,
www.morphdbase.de/?J_Krieger_20170807-M-67.1,
www.morphdbase.de/?J_Krieger_20170807-M-66.1,
www.morphdbase.de/?J_Krieger_20170807-M-65.1,
www.morphdbase.de/?J_Krieger_20170807-M-64.1,
www.morphdbase.de/?J_Krieger_20170807-M-63.1,
www.morphdbase.de/?J_Krieger_20170807-M-62.1,
www.morphdbase.de/?J_Krieger_20170807-M-61.1,
www.morphdbase.de/?J_Krieger_20170807-M-60.1,
www.morphdbase.de/?J_Krieger_20170807-M-59.1,
www.morphdbase.de/?J_Krieger_20170807-M-58.1,
www.morphdbase.de/?J_Krieger_20170807-M-57.1,
www.morphdbase.de/?J_Krieger_20170807-M-56.1,
www.morphdbase.de/?J_Krieger_20170807-M-55.1,
www.morphdbase.de/?J_Krieger_20170807-M-54.1,
www.morphdbase.de/?J_Krieger_20170807-M-53.1,
www.morphdbase.de/?J_Krieger_20170807-M-52.1.

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
