# Peer review of "Evaluation of standard imaging techniques and volumetric preservation of nervous tissue in genetically identical offspring of the crayfish Procambarus fallax cf. virginalis (Marmorkrebs)"

_PeerJ, doi:10.7717/peerj.5181_

## Round 0.1 · original submission · Major Revisions

I apologize for the delay in making my decision, but I was waiting for an additional review which never came. To avoid, further delay I decided to based my decision on my own assesment and other 2 reviews. Your study is a valuable contribution to the field of microscopic volumetric analysis of biological soft tissue samples based on 3D approaches. However, there would be some crucial points to integrate to make your study of even greater value and more generally comparable.

The main points are:

Experimental design: The conclusions which are drawn cannot be fully made without the current experimental design (see comments by reviewer 2). I would suggest to do additional measurements (see comments by reviewer 2) or change the title and rewrite part of the manuscript to encompass conclusions which can be made with your set-up (compare also reviewer 1). Your title and the manuscript should also be changed/rewritten accordingly.

Lack of in-vivo measurements: both reviewers stated that in vivo data would need to be generated to make certain statements and make your data fully comparable.

Reported measurements: it is great that you made your original data available through morphodbase. However, the manuscript would benefit from adding sample sizes to boxplots and providing raw measurements of all specimens in a table (see comment by reviewer 1).

Formatting: your manuscript could be restructured a bit to make it easier to follow (see suggestions by reviewer 1).

In addition, to taking into these points and other suggestions, please also address.

Line 227: statistics – please list package/functions in R for particular statistical test as well as references introducing these test

Figure 2: please provide sample size (n) for each category for completeness sake

I choose major revisions because the reviewers suggested additional analyses or major restructuring of the conclusions. However, I believe that these changes are worthwhile to consider and can be fairly easily be made. I am looking forward to publication of this manuscript after revision.

Reviewer 1 ·

Basic reporting

See "general comments for the authors".

Experimental design

See "general comments for the authors".

Validity of the findings

See "general comments for the authors".

Additional comments

In their manuscript “Volumetric evaluation of standard fixatives and imaging techniques on nervous system preservation in genetically identical offspring of the crayfish Procambarus fallax cf. virginalis (Marmorkrebs)” Emanuel S Nischik and Jakob Krieger compare the impact of different fixation methods on the tissue preservation of a crustacean brain. The authors use the uprising model-organism Procambarus fallax cf. virginalis, which reproduces by parthenogenesis and thus produces genetically identical offspring. Three fixation methods and two imaging techniques are investigated, all cutting edge techniques, not only in the (neuro-)morphological community. The authors concentrate their investigation on the volumetric changes of three distinct brain areas, the DCL, the AcN, and the midbrain. Nischik and Krieger demonstrate that µCT procedure is the least disruptive method, whereas fixation in Karnovsky solution leads to a volume decrease of 24 %, and Zinc-formaldehyde fixation to a decrease of 19%. This is the first investigation generating basic data to obtain conversion factors in order to allow interspecific comparisons of volumetric data and thus represents an important contribution to the literature.
Overall, the study is well executed and the datasets provided are convincing and well supported by the 5 figure plates and one table. The raw data of all scanning series can easily be accessed via MorphDBase as described in the manuscript. The introduction shows the context of this research and interpretations in the discussion are well supported by the presented datasets. However, the text is often confusing and seems unstructured (see major issues point 1). Additionally, I have several other major issues. If these and the minor points below are addressed I would definitely recommend this paper for publication in PeerJ.
Major Issues:
1.) The paper needs some structural modifications. In the following I will give some examples, but the authors should check the whole manuscript for possible modifications.
a) The abstract could be more effective in clarifying the purpose of the paper. In my opinion some sentences concerning an introduction or motive for the study might be necessary at the beginning. It should be mentioned that volumetric data is an important tool in neuroanatomy. However, due to the application of different methods these data are hardly comparable and that this account starts the first approach for a posteriori comparison of already existing data in the literature (it is nicely described in the introduction).
b) Results: The results start with a subheading “MicroCT” (line 238). If I understand it right, in the second part of this subheading the authors present volumetric data of all three methods applied (Line 248, starting with …”While the volume of the accessory lobes…”). However, this should be done in the section where all methods are compared, e.g., lines 276, section MicroCT versus cLSM. And actually, the same data are already compared here, namely the volume of left and right hemispheres of paired lobes. Compare:
Lines 249-251: “…no difference in the volume of the left and right hemispheres of paired lobes could be found in any given treatment (paired Student’s t-test: DCL: p = 0.2691; AcN: p = 0.1107; n = 20).
Lines 288-289: “No significant differences for left and right lobes (paired Student’s t-test: DCL: p = 0.2691; AcN: p = 0.1107; n = 20) have been detected.”
c) Discussion: In Lines 361-364, the authors discuss that µCT-preparations feature the closest in vivo coherence, but are influenced by the fixation and dehydration leading to shrinkage. While reading the manuscript for the first time, at this point I was wondering why no in vivo data have been generated. The discussion of in vivo methods however follows on the next page, starting with line 391, too late in my opinion. It might help shifting the whole paragraph to line 364.

2.) Lines 69-71: The authors mention several aspects of tissue processing, which influence its preservation. I think they missed the factor temperature at which the fixative is applied, as it influences the speed of penetration. In this context, I noticed that this information is sometimes missing in the M&M section (e.g., lines 168/169 or 185-188). The authors should mention all relevant settings in the M&M section, in order to allow other scientists to repeat their experiments. This is especially important regarding the scope of this paper (comparison/evaluation of volumetric data/methods). For example, in line 147 it is not clear if the 3 steps of 99.5% ethanol are incubated for 30 min as well, and the concentration of NaN3 is missing (line 186). Furthermore, supplier of reagents are not given consistently, e.g., supplier of the Synorf-Antibody is missing.

3.) In the context of the previous issue, please indicate in the M&M section how many specimens were used for each experiment. This information is given for µCT, but not for the other cLSM studies. This information should also be included in the Figures 1 and 2.

4.) The authors make great effort measuring volumetric data of three different brain regions. However, they do not show the raw data. As the “n” is not too large they should add a Table with their results for each brain. In the same context, the authors have to be aware that the data are based on a small “n”. That should be taken into account when discussing the results and statistics…

5.) Lines 277/278 (see also 297-299): The authors state that all three tested methods differ to the in vivo congruent. But this is not documented by their volumetric data, as in vivo measurements have not been performed. The authors should rephrase this statement into a more careful and hypothetical sentence. It is just an assumption made due to the reasons explained in the next sentences. By the way, in my opinion, these explaining sentences (lines 278-281: Starting with: …. Since all samples at least temporarily….) are part of the discussion.

6.) The intriguing idea of this paper is “to obtain conversion factors in order to deduce approximate in vivo volumina” (lines 414-415) in order to “allow for a posteriori standardization of determined nervous tissue volumes in malacostracans” (lines 416-417). The authors mention some works from the literature, where volumetric data concerning the DCLs are given (Table 1). My question is: can the authors apply their conversion factors (percentage of shrinkage) to the cited studies? I guess it is still not possible to apply these conversion factors, as there are too many variables (fixation time, temperature, etc) and the authors should make clear that this study has to be interpreted as a pioneering study.

7.) In the same context, an often applied fixation method is 4% PFA for several hours or overnight as done in the cited literature (Beltz et al., 2003; Krieger et al., 2010). Would it not be logically consistent to obtain a conversion factor for this methodological setup (I assume that fixation in Karnovsky versus PFA might result in different shrinkage levels)? Could you explain why you have not conducted a fourth experiment with a fixation in solely 4% PFA?

8.) Does the entire nervous tissue shows the same degree of shrinkage or do different areas/neuropils show different degrees of shrinkage? This should be made clear to the reader, as a putative conversion factor might not apply for all parts of the brain.

9.) I am wondering if the fact that the brain for the µCT is incubated in fixative solution without further dissection prevents shrinkage. Maybe the surrounding tissue supports a more careful preservation of nervous tissue? Do the authors have data concerning Karnovsky- or ZnFA-fixed brains inside the head capsule without dissection?


Minor Issues:
1.) In-text citations concerning three-author-papers are inconsistent: Sometimes they are abbreviated (e.g. line 60: Akkari et al., 2015), sometimes all authors are listed (e.g. line 341: Durst, Eichmüller & Menzel, 1994).

2.) I am wondering why the mixture of glutaraldehyde and PFA is not termed Karnovsky’s solution throughout the manuscript, as done for Bouin’s solution.

3.) Line 36: Please replace “Zinc-Formaldehyde” by “zinc-formaldehyde”

4.) Line 60/61: Please rearrange order of references chronologically (the same issue applies for line 91)

5.) M&M section: Please replace “second antibody” by “secondary antibody” throughout the manuscript.

6.) Line 133: Please add a period at the end of the section.

7.) Line 193: Please replace “3c11” by “3C11”.

8.) Line 207: Please replace “Cy3-conjugats” by “Cy3-conjugates”.

9.) Line 267: Please replace “statically” by “statistically”.

10.) Line 273: The cross reference to Fig. 3D is wrong. It should by Fig. 4D.

11.) References: Some references are cited with the DOI, several are not.

12.) Legend of Table 1: Please add the abbreviation of “deutocerebral chemosensory lobe” (DCL).

13.) Table 1: The correct citation for Birgus latro is “Krieger et al., 2010”.

14.) Table 1: Please indicate if the mean of DCL volume for Procambarus is the mean of all three methods used. And would it not be more appropriate to separate between the three fixation methods considering that there is a significant difference at least between two methods (Figure 3)?

15.) Legend of Figure 1: “accessory lobe” should be changed to the plural form.

16.) Legend of Figure 2: The abbreviations DCLs and AcNs should be introduced when the corresponding term is used first.

17.) Figure 4: Some lettering is hardly readable in this Figure (for example CA, TN, 9/11, 10 in A, and 9/11 in B). Maybe a contrasting contour might help here.

18.) Legend of Figure 5: Explanations of the following abbreviations used in the Figure are missing in the legend: AIINv, LAN.

Reviewer 2 ·

Basic reporting

no comment

Experimental design

no comment

Validity of the findings

In my opinion some of the conclusions cannot be drawn based on the current experimental design. This point will be elaborated in detail in the "general comments to the author" section. I make some suggestions on how to re-phrase the conclusions and how to sharpen the interpretation of the findings.

Additional comments

The manuscript “Volumetric evaluation of standard fixatives and imaging techniques on nervous system preservation
in genetically identical offspring of the crayfish Procambarus fallax cf. virginalis (Marmorkrebs)” by Nischik and Krieger is a valuable contribution to the field of microscopic volumetric analysis of biological soft tissue samples based on 3d images. It provides a comparison of image quality and sample shrinkage between standard preparation protocols for two of the most widely used microscopic 3d imaging techniques: confocal microscopy and x-ray microCT. Evaluation is based on measurements obtained from different parts of the central nervous system (brain) of the crayfish Procambarus fallax cf. virginalis. While the intention of the study is clear, the conclusions drawn by the authors suffer from some severe technical shortcomings based on the study design. First, the experimental design does not allow evaluating differences in tissue shrinkage based on different fixatives (see major concerns, point 1). Second, no measure of in vivo brain volume is provided, so there is no estimation of absolute shrinkage (see major concerns, point 2).

Despite these shortcomings, several aspects of this study are positive and should be highlighted:
- The quality of the image data is very good.
- I highly appreciate the fact that all data were deposited at the online repository MorphDBase! Making the original image data available is great for several reasons! It also helped me with interpretation of the data while doing this review.
- The presented data highlights some characteristics of the used tissue preparation protocols and imaging techniques. It shows that the tested protocols provide good image contrast.
- The data also shows that microCT imaging is much more suitable for doing volumetry than confocal microscopy, because several technical limitations yield an anisometric image quality in optical sectioning. This is very obvious when looking at the original image volumes (see major concerns, point 4).

Taken together the positive aspects and the technical shortcomings, I suggest to make clear that…
- the present experiment does not allow to make some statement about the contribution of the used fixative to total specimen shrinkage
- no attempt was made to use or develop a protocol that minimizes specimen shrinkage (see major concerns, point 3)
- the present data only allows for an evaluation of currently used standard protocols in terms of final specimen size and image quality
- one of the main findings of this study is the difference in image volume quality (isotropy) between confocal and microCT images

Further I have one more suggestion: The authors should consider scanning some (5-10) fresh brains after dissection with microCT (see major concerns, point 2a) and calculate the average volume of a fresh brain (central brain) at the investigated developmental stage. This would yield two major benefits. First it would allow calculating absolute shrinkage values for the tested protocols. Second, this data would provide a baseline for future studies for refining fixation and staining procedures in order to minimize shrinkage.
I know that asking for additional experiments during a revision is somehow problematic. The authors would have to have immediate access to (i) fresh specimens and (ii) the microCT scanner. In the present case I think that this requirements could be fulfilled. If not, the authors can state why they are not able to do it. The scan resolution does not need to be very high (e.g. acquired with a detector binning of 4) so the time for a tomography could be roughly 15 minutes. This would allow the authors to collect data from 10 brains within one or two days (dissection and scanning). With maybe 2 days for image segmentation, volumetric data for whole brain volume could be generated within one week. Personally I think that this data would be absolutely valuable and would dramatically increase the scientific value of this manuscript.

Despite my criticism, I think that after a serious revision the paper would be a valuable contribution to your journal. Before publication I ask the authors to consider the following more detailed comments when revising the manuscript.


Major concerns:

-1- This is my main criticism. In the abstract the authors state that “Fixation in paraformaldehyde of dissected brains lead to a volume decrease of 36 24 %, whereas fixation in Zinc-Formaldehyde resulted in a shrinkage of 19 %.” (referee comment: this decrease is calculated relative to the volumes measured on Bouin’s material scanned with microCT, not as a decrease from in vivo tissue volumes). However, the measured differences in volumes reflect shrinkage occurring during the whole sample preparation protocol, and not only during fixation. Using the present experimental design, an evaluation of different fixatives concerning volume shrinkage cannot be done. For doing this, a much simpler experimental design is needed: testing a couple of fixatives and keeping the rest of the workflow (intact head specimen vs. dissected brain, washing steps, staining, clearing, mounting, etc.) constant. In the present experiment multiple variables are present. Many of the steps very likely introduce some tissue shrinkage, and just from the final volume measurements it is not possible to say which steps contributes most to the shrinkage finally observed in the image volume. Thus no statement about the contribution of fixative to shrinkage is possible.
Several more detailed comments should underline this statement:
-a- dehydration to absolute ethanol as used in the microCT protocol is known to introduce tissue shrinkage (traditional histological paraffin wax embedding protocols...)
-b- staining with iodine introduces shrinkage due to changes in osmolarity as proven before (see for example Buytert et al 2014, Microscopy and Microanalysis)
-c- clearing with methyl salicate introduces shrinkage as well (see for example Bucher et al 2000, J Neuroscience Methods)
-d- microCT images were acquired from intact heads, confocal images were acquired from dissected brain specimens. Besides possible artefacts coming from the dissection procedure, the dissected brains might respond differently to chemicals and finally the mounting of dissected brains for confocal microscopy might introduce some further mechanical distortion

Taken together these facts, the authors should consider to re-phrase the title and to re-structure the presentation of their findings. What the authors do show is that the applied standard protocols yield strong differences in tissue preservation, shrinkage, and volume image quality. What the authors don´t show is how the fixatives specifically contribute to these findings.

-2- The second main criticism is that there is no measurement of absolute tissue shrinkage, as no data on fresh tissue volumes before fixation is provided. This point is well aware to authors and they consider “direct volumetry using Archimede´s principle or high resolution MRI” (page 19, line 399) as promising approaches to measure in vivo brain volumes. In principle I agree, but in practice I think that direct volumetry using Archimede´s principle is very hard to do with such small volumes and might be prone to measurement errors, and MRI might fail to provide sufficient resolution on native tissues (without use of paramagnetic contrast agents). The authors should consider microCT as an alternative method to measure in vivo or at least fresh (unfixed) total brain volume. Two approaches seem feasible:
-a- Fresh dissected brains could be mounted in conical containers (e.g. pipette tips) in air. Some water/buffer is added at the bottom of the tube and the tube is tightly sealed. This will provide a moist atmosphere and prevent drying of the specimen. A quick tomography can be done just to see the boundary of the brain. Most scanners will do such scans in 10-15minutes by using detector binning (if you plan to further process the sample, the brain would be fixed after the scan; the overall time from dissection to scanning to fixation is still below 30min, which might be acceptable for many kinds of analysis).
-b- Alternatively the authors could consider scanning intact heads of freshly killed specimens with in-line phase contrast. This might give sufficient contrast to discriminate between boundaries on brain to surrounding tissues, although it might be hard to do this scans quick enough with laboratory microCT setups (they could however be done at synchrotron beamlines).

-3- I understand the point of using “standard” protocols for estimating tissue shrinkage, as the overall goal of this study is to make volume measurements comparable even between different preparation protocols and imaging modalities. The presented protocols (fixatives, contrast agents) have been utilized in the past in many studies and have proven to provide excellent tissue preservation and image contrast for different kinds of qualitative morphological evaluations. Despite this fact, I doubt that the presented protocols are suitable for avoiding volume shrinkage. I will elaborate this point on one example: the microCT protocol including Bouin´s fixation and iodine staining. While iodine staining on Bouin´s fixed material has yielded excellent qualitative morphological data in the past (Sombke et al 2015, J Comparative Neurology), I don´t think that the protocol presented in the present paper is the best choice for reducing tissue shrinkage artefacts. This is for three main reasons:
- a- Previous quantitative studies did show that tissue shrinkage and tissue deformation in Bouin’s fixed tissue is quite severe (Schmid et al 2010, BMC Developmental Biology) and worse compared to other fixatives.
-b- Dehydration to 100% ethanol
-c- Using 2% Iodine is not a good idea in my opinion. Many studies in the past used 1% elemental iodine (Metscher 2009, Developmental Dynamics; Sombke et al 2015, J Comparative Neurology). The concentration of the contrast agent will affect osmolarity, and higher concentrations will cause more shrinkage.
Instead, other protocols could be considered. For example, Wong et al (2013, PLOSOne) showed that shrinkage in staining with Lugol´s iodine potassium iodide solution could be reduced to a minimum by embedding samples in a hydrogel mesh before staining. Given the data in the literature that already addressed these issues, I think the authors should re-phrase their main argument.

-4- Different imaging techniques have different kinds of bias and physical limitations. This is a general shortcoming that makes it very hard to compare microCT data with confocal data. For microCT data we can assume that image geometry is correct. For confocal data we cannot simply assume that, based on somewhat limited z-resolution depending on the PSF of the system, a possible refractive index mismatch, …

I looked into the original image stacks deposited at MorphDBase with some detail. The quality of the images is good compared to recent standards in microCT and confocal imaging! Still, if you reslice these stacks in in XZ or YZ plane, the geometric issues with confocal data become very well visible. While the brains have a very “nice 3d structure” in the microCT scans, they look much more “flat and squeezed” in the confocal images. Many factors may contribute to this phenomenon, including dissection, squeezing during mounting, optical artefacts, and others.
While this makes it even harder to evaluate contribution of different chemical procedures to tissue shrinkage it obviously points towards another fact: tomographic techniques (like microCT) are highly superior compared to optical sectioning techniques (like confocal microscopy) regarding geometric image isotropy. This means that tomographic techniques are highly superior for tissue volumetry, despite current limitations in image resolution and specific labelling techniques.

Minor concerns:

-5- I don’t understating the difference between general sample preparation (Line 135) and sample preparation for microCT / cLSM. What does general mean? Does It mean that every specimen was fixed in 4% PFA before being fixed in Bouins?

-6- Please check the consistency in wording. One example is the alternating use of microCT and µCT throughout the whole text. Please use one or the other, but not both.

-7- Page 12 Line 206-209 Which confocal microscope was used? Company/model? Inverted or upright? Pinhole size/section thickness? Slice interval? This is relevant for interpretation of image data and volume measurements.

Some more suggestions on wording:

Page 9 line 125 Change “Juveniles started feeding individually” to “juveniles started autonomous feeding”

Page 11 Line 188 Rephrase “abundant primary antibody” by “excess primary antibody”

Page 12 Line 217 The material statistics tool uses the voxel data and not the surface to calculate volumes

Page 12 Line 219 Remove information on threshold, as no explanation is provided for image intensities (8 bit/16bit, byte scaling during reconstruction, calibration, beam hardening correction,…)

Page 16 Line 320 Resolution and image noise do not necessarily correlate. Resolution is limited by the microCT scanner (minimal spot size, specifications of the scintillator and optical detector…) while using longer exposure times can increase the SNR.

---

## Round 0.2 · Minor Revisions

Thank you for addressing our suggestions and doing additional experiments. The manuscript has become even more interesting and easier to follow. I feel the manuscript is in good state and I look forward to publish the results of your experiments. There are just some additional minor, but crucial points I would like to address before publication:

Use of adults for additional experiments: I greatly appreciate the additional analyses you performed, which make your study even more important. I guess this might have been driven by availability, but you at least need to explain why you used adults and how this could potentially have effected the results (see also comments by reviewer 2). You also need to rewrite some parts in the text where you state you only used juveniles, by saying you focusing on juveniles without some exceptions (see my comments in the annotated pdf).

Use of abbreviations: Please make sure the use of abbreviations is consistent throughout the manuscript. They should be defined before their first use (e.g., in methods) and used consistently after that. They could of course also be written in full before their definition (e.g., in the introduction) and common use (see comments by reviewer 1).

Treatments of optical artefacts in confocal microscopy: There are still some issues with estimating the refractive index mismatch (see comments by reviewer 3). Reviewer 3 suggested three potential ways to resolve this issue.

Supplementary material: I agree with reviewer 1, that I would be ok to add supplementary table in the text and that might make it easier to follow in this case. Please consider it if it make sense to you.

Please address all reviewer suggestions and my suggestions in the annotated pdf in addition to these points.

Reviewer 1 ·

Basic reporting

In this revised version, the authors have made a serious effort to modify their manuscript and responded to all issues of prior reviews. In particular, they have streamlined the text, added important information as well as supplementary data, and addressed the major criticism concerning missing in vivo data by a more detailed discussion and additional experiments.
Therefore, I strongly recommend this revised manuscript for publication in PeerJ.
I have some very minor suggestions, which could further improve the manuscript (line numbers refer to the clean version without tracked changes):
1.) I would prefer including the supplementary tables into the main body of the manuscript as they contain important data and the journal is not restricted by page limitation. This will be far more comfortable to the readers. However, I leave this decision over for the editor.
2.) Please check introduction of used abbreviations. For example, in line 175 “room temperature” is abbreviated by RT, but written out throughout the rest of the text. HEPES buffered saline and zinc-formaldehyde are abbreviated first in lines 201/202, but are already mentioned before (just two examples, there might be more issues).
3.) Line 148: Change Boiun’s to Bouin’s.
4.) Line 334: ‘thevc’ should be changed to whatever the authors meant to write here.

Experimental design

see above

Validity of the findings

see above

Reviewer 2 ·

Basic reporting

no comment

Experimental design

no comment

Validity of the findings

Some findings must be re-interpreted after considering an appropriate method for correcting z-axial scaling in confocal images.

Additional comments

In the revision of the manuscript “Evaluation of standard imaging techniques and volumetric preservation of nervous tissue in genetically identical offspring of the crayfish Procambarus fallax cf. virginalis (Marmorkrebs)” by Nischik and Krieger, the authors managed to resolve most of the concerns raised by the two reviewers. I highly appreciate the fact that the authors carried out an additional experiment for measuring fresh brain volumes and for estimating tissue shrinkage before and after Bouin´s fixation and microCT staining.

However, one main criticism was not yet resolved. This issue concerns the topic “optical artefacts in confocal microscopy” and has to be addressed before the paper is ready for publication. This topic is of central importance to this manuscript, thus in my opinion justifying a second major revision.

In my review of the initial submission I referred to this point as “Major concern 4”. I criticized the fact that optical sectioning techniques produce different artefacts compared to µCT, related to shrinkage, specimen squeezing, refractive index mismatch, etc. The authors tried to resolve this topic in the revised manuscript, but in opinion they do not treat the topic “refractive index mismatch” adequately. They recognize this problem and try to correct for z-axial scaling in the revised paper (conversion factors, supplementary tables), as seen in some passages of the revised text, such as:

Lines 412- 417 “However, using the correction factor for the refractive mismatch for the use of methyl salicylate provided by Bucher et al. (2010) of 1.581 (for a 10x dry objective with a numerical aperture of 0.40) resulted in a drawn-out appearance of the brains and led to an aberrant ratio between the dimensions in xy-axis and z-axis. By manually adjusting the voxel z-dimension of confocal image stacks by a factor of 1.188, led to spatial congruence of the brain dimensions of µCT-scans”

However, I argue that “manually estimating” a correction factor is not a scientifically correct approach. How was this congruence with µCT measured? Based on physical principles the correction factor for the present study should be somewhat close to the number calculated by Bucher et al (identical media = air/methyl salicylate, identical magnification = 10X, identical numerical aperture = 0.4). I also checked one of the morphdbase datasets (M64-001.1) and to me a correction factor of 1.58 looks quite plausible on this dataset.
I have three suggestions how to resolve this issue:
-a- Calculate a theoretical correction factor using one of the established approaches (Hell et al 1993, Journal of Microscopy 169:3, 391-405)
-b- Practically measure the correction factor with some PMMA spheres under identical conditions as in brain measurements (Besseling et al 2015, Journal of Microscopy 257:2, 142-150)
-c- Use the values from Bucher et al as an approximation, which is in my opinion acceptable for reasons mentioned above (identical media, identical magnification, identical numerical aperture).

However, if the z-axial correction factor is really higher then 1.5, the main conclusion resulting from this is the following: the two tested confocal preparation protocols shrink the sample less than the tested microCT preparation protocol. This finding would need to be embedded into the manuscript. The whole paper is about volume measurements, so it needs a very serious treatment of that issue. The discussion could be somehow structure like this:
-i- These are the volume measurements I get from the uncorrected image volumes (µCt measurements are larger than confocal measurements)
(ii) This is for several reasons, including sample prep artefacts, imaging artefact etc.
(iii) Confocal data can be corrected for refractive index mismatch using different approaches
(iv) These are the volume measurements I get after correcting confocal volumes
(v) Finally discuss all the strengths and pitfalls of the different preparation and imaging protocols

Despite this major concern I have one more minor comment: the authors should state clearly why they did not use same aged individuals for measuring fresh brain volumes. While the main analysis was done on juvenile brains, adult animals were used for measuring fresh volumes. Brain volumes of adults are roughly 100 times larger compared to the juvenile animals. This could be a problem, because the chemical composition (e.g. protein or lipid content) of adult brains could be different, which would mean that they might also respond differently to fixation, dehydration, etc.

---

## Round 0.3 · Minor Revisions

Thank you for describing the treatment of optical arefacts in greater detail, integrating the supplementary table in the text, homogenizing the abbrevations as well as addressing all other points raised by the reviewers. The manuscript is now also easier to follow now that you explain the rationale behind using adult instead of juveniles in greater detail. Your manuscript is as good as accepted, i just had one more thing i want you to take care before publication.

You mention changing EU regulations concerning trade of the marbled crayfish. It would be appropriate in general to mention how you and which EU/national legislation you followed when euthanizing the animals in your experiments for completeness sake.

---

## Round 0.4 · accepted · Accept

Thank you for integrating into your manuscript that animals were euthanised according to EU legislation.

#